# Explainable ensemble learning method for OCT detection with transfer learning

**Jiasheng Yang**[1☯]*, **Guanfang Wang**[2,3☯], **Xu Xiao**[4], **Meihua Bao**[1], **Geng Tian**[3]*

**1** Academician Workstation, Changsha Medical University, Changsha, Hunan, China, **2** School of Mathematics and Statistics, Nanjing University of Information Science and Technology, Nanjing, Jiangsu, China, **3** Geneis Beijing Co. Ltd., Beijing, China, **4** School of International Education, Anhui University of Technology, Maanshan, Anhui, China

☯ These authors contributed equally to this work.
* jsyang.mcc@gmail.com (JY); tiang@geneis.cn (GT)

**Data Availability Statement:** Data relevant to this study are available from https://www.kaggle.com/datasets/jasonyangjs/oct-dataset.

**Funding:** This study was funded by the Hunan Key Laboratory of the Research and Development of

## Abstract

The accuracy and interpretability of artificial intelligence (AI) are crucial for the advancement of optical coherence tomography (OCT) image detection, as it can greatly reduce the manual labor required by clinicians. By prioritizing these aspects during development and application, we can make significant progress towards streamlining the clinical workflow. In this paper, we propose an explainable ensemble approach that utilizes transfer learning to detect fundus lesion diseases through OCT imaging. Our study utilized a publicly available OCT dataset consisting of normal subjects, patients with dry age-related macular degeneration (AMD), and patients with diabetic macular edema (DME), each with 15 samples. The impact of pre-trained weights on the performance of individual networks was first compared, and then these networks were ensemble using majority soft polling. Finally, the features learned by the networks were visualized using Grad-CAM and CAM. The use of pre-trained ImageNet weights improved the performance from 68.17% to 92.89%. The ensemble model consisting of the three CNN models with pre-trained parameters loaded performed best, correctly distinguishing between AMD patients, DME patients and normal subjects 100% of the time. Visualization results showed that Grad-CAM could display the lesion area more accurately. It is demonstrated that the proposed approach could have good performance of both accuracy and interpretability in retinal OCT image detection.

## Introduction

Age-related macular degeneration (AMD) and diabetic macular edema (DME) are prevalent eye conditions that are becoming more common among elderly individuals and those with diabetes globally. These conditions can result in progressive vision loss and, in severe cases, even blindness. With the global aging population, increased usage of electronic devices, and growing numbers of diabetic patients, the prevalence of DME is on the rise [1, 2]. As the leading causes of irreversible vision loss worldwide, AMD and DME require reliable and accurate diagnostic and monitoring tools [3–6]. Optical coherence tomography (OCT) has become a critical imaging technique for diagnosing and tracking patients with AMD and DME, offering

Novel Pharmaceutical Preparations, The Hunan Provincial Key Laboratory of the TCM Agricultural Biogenomics, and the "Double-First Class" Application Characteristic Discipline of Hunan Province (Pharmaceutical Science) in the form of salaries to MB. This study was also funded by the Foundation of Hunan Educational Committee in the form of a grant to MB [23A0662].

**Competing interests:** I have read the journal's policy and the authors of this manuscript have the following competing interests: The authors declare that they have no known competing financial interests or personal relationships that could have appeared to influence the work reported in this paper. This does not alter our adherence to PLOS ONE policies on sharing data and materials.

a noninvasive and touch-free solution. Its application could help healthcare professionals detect these conditions early and initiate prompt and effective treatment plans, potentially improving patient outcomes and quality of life [7–9].

OCT images are useful for diagnosing two common retinal pathologies, AMD and DME. In AMD, the presence of drusen, changes in the retinal pigment epithelium, and the buildup of subretinal and intraretinal fluid are key OCT findings. DME, on the other hand, is characterized mainly by hard exudates or thickening of the center of the macula. In both cases, the affected retina appears markedly different from that of a healthy individual [10]. Traditionally, ophthalmologists have had to manually examine each cross-section of an OCT volume, resulting in a significant increase in workload and limited capacity to obtain accurate diagnoses [11–13]. As a result, the need for automated identification of retinal pathologies through the analysis of OCT images has become increasingly apparent.

Several automatic detection technologies that use OCT images have been developed to identify specific retinal pathologies through either segmentation or classification. Ebrahim Nasr Esfahani et al. [14] introduced a new edge convolutional layer (ECL) that accurately extracts retinal boundaries in various sizes and angles with fewer parameters than the conventional convolutional layer. Using this layer, they proposed the ECL-guided convolutional neural network (ECL-CNN) method for automatic OCT image classification, achieving an average precision of 99.43% for a three-class classification task. Anju Thomas et al. [15] proposed an efficient algorithm for detecting AMD from OCT images using a multiscale and multipath convolutional neural network architecture. Their proposed CNN includes multiscale and multipath Convolutional Layers (CL), and they tested the combination of three datasets, achieving an accuracy of 0.9902. Maidina Nabijiang et al. [16] proposed a novel attention mechanism called the block attention mechanism, which actively explores the role of attention mechanisms in recognizing retinopathy features. The experimental results showed that the proposed framework outperformed existing attention-based baselines on two public retina datasets, OCT2017 and SD-OCT, achieving accuracy rates of 99.64% and 96.54%, respectively. However, these deep learning methods either required a large amount of training data and weeks to achieve a classification accuracy when training the CNN from scratch using raw images, or resulted in poor performance of classification when using the feature-based transfer learning method without further optimization and additional fine-tuning.

To overcome these limitations, machine learning techniques can be utilized to learn representations through pre-training and have been employed for automatic classification of medical images [17–20]. In a study by Feng Li et al. [21], a deep transfer learning method was employed to fine-tune the ResNet network pre-trained on the ImageNet dataset. The performance of the approach was evaluated on a validation dataset, and metrics such as prediction receiver-operating characteristic (ROC), sensitivity, accuracy, and specificity were computed. The experimental results demonstrated the superior performance of the proposed approach in detecting retinal OCT images, achieving a prediction sensitivity 97.8%, accuracy 98.6%, specificity 99.4%, and introducing an area under the ROC curve of 100%. Similarly, KARRI et al. [22] fine-tuned a pre-trained convolutional CNN, GoogLeNet, to improve its prediction capability and identify salient responses during prediction to understand learned filter characteristics. The fine-tuned CNN effectively identified pathologies compared to classical learning. Another proposed method by SamanSotoudeh-Paima et al. [23] introduced a multi-scale CNN based on the feature pyramid network (FPN) structure. Pre-trained ImageNet weights were used to enhance the performance of the model from 87.2% ± 2.5% to 92.0% ± 1.6%. Although the classification methods discussed above achieved promising results, their generalization capability to other fields is limited.

The aforementioned recognition methods all utilize a single deep learning framework, which offers greater flexibility due to its non-linear approach [24–26]. However, this flexibility can also result in higher variance due to the randomized training algorithms used, which can lead to different prediction results and make it challenging to develop a final model for prediction. To address this issue, ensemble learning can be used, which involves training multiple models and combining their predictions to reduce variance and improve overall accuracy [27–29]. In fact, ensemble learning can often produce better predictions than any single model [30]. For instance, Ashok et al. [31] proposed two computer-aided diagnosis (CAD) methods that utilize ensemble deep learning models based on Inception-V3 and ResNet to classify OCT images into four categories: normal, choroidal neovascularization (CNV), vitreous warts, and DME. Similarly, Mousa Moradi et al. [32] demonstrated that stack-based ensemble deep learning can enhance the detection of both non-advanced and advanced AMD by comparing the classification results of the base model and the ensemble model. Ai et al. [33] introduced a novel global attention block (GAB) that enhances classification performance when integrated with any CNN, resulting in a notable improvement of 3.7% accuracy compared with the EfficientNetV2B3 model. Huang et al. [34] proposed FN-OCT, a fusion network-based algorithm for retinal OCT classification that fuses predictions from InceptionV3, Inception-ResNet, and Xception classifiers with CBAM, employing three fusion strategies. On the dataset from UCSD, FN-OCT achieved 98.7% accuracy and 99.1% AUC, surpassing InceptionV3 by 5.3%. Akinniyi et al. [35] proposed a multi-stage classification network using OCT images for retinal image classification. Their architecture utilizes a pyramidal feature ensemble built on DenseNet for extracting multi-scale features. Evaluation on two datasets demonstrates advantages, achieving high accuracies for binary, three-class, and four-class classification. Kayadibi et al. [36] proposed a hybrid approach for retinal disease identification. It uses a fully dense fusion neural network (FD-CNN) and dual preprocessing to reduce speckle noise, extract features, and generate heat maps for diagnosis confidence.

In this paper, our most prominent contribution is the development of an explainable transfer ensemble model to effectively identify age-related fundus macular degeneration, which is an ensemble architecture based on migration learning. The proposed ensemble architecture helps the model to reduce the variance in predictions compared to a single independent model. We conducted various experiments to optimize the model and evaluate its performance using standard evaluation metrics. We investigated the effect of pre-training on the network by comparing experiments on the original and weighted networks. We also conducted experiments to study the performance of the model with different combinations of sub-networks. Finally, this paper illustrates the learning content of the model through Class activation mapping (CAM) as well as gradient—weighted Class Activation Mapping (Grad-CAM) visualization. The following section provides detailed description of the proposed method. Section 3 discusses various experiments performed to evaluate the proposed method. Section 4 discusses the paper. Finally, section 5 concludes our work.

## Materials and methods

In this section, we propose an explainable deep ensemble learning model based on migration learning, which has three basic models and a prediction block as shown in Fig 1. The details are described in the following subsections.

### OCT image data

In this study, we designed and evaluated the proposed algorithm using a publicly available dataset [37]. The dataset consisted of normal, dry AMD, and DME categories, each of which

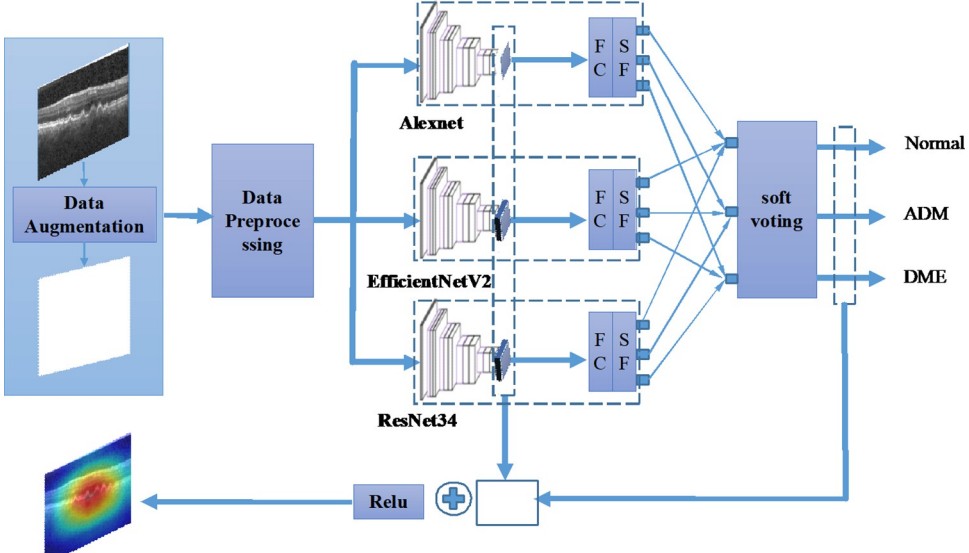

**Fig 1. Organizational structure of the proposed explainable ensemble deep learning method for OCT image detection.**

included 15 subjects with multiple images per subject. The data list is shown in the Table 1, and Fig 2 displays example from each class.

To facilitate downstream work, we pre-process the images. Its general operations include image noise removal, image quality improvement, image resizing, data enhancement, histogram equalization, contrast processing etc. However, in order to avoid overfitting and improve the generalization ability of the classifier, we exclude operations such as image denoising and equalization. In this paper, the image preprocessing process involves data enhancement through horizontal and vertical flip as well as rotation within a range of -15˚ to 15˚. Then the image is normalized. Finally, the image is normalized to a uniform size of 224*224 pixels.

## Pre-trained CNN models

In this section, we will provide details on the CNN models utilized in this study and the corresponding training approach. To select the most appropriate architecture, we assessed various convolutional CNN models, including Alexnet, Efficientnet_v2, and Resnet34. Alexnet [38–40] consists of a total of eight layers, which comprise of five convolutional layers and three fully connected layers. It deploys rectified linear units (ReLU) in lieu of tanh functions and overlapping pooling to prevent overfitting during training. ResNet-34 is a well-known model within the deep residual learning framework (ResNet) [41], representing a relatively shallow network. It is primarily composed of 16 basic units, one product layer, and one fully connected layer, totaling 34 layers. Efficientnet_v2 [34] is a family of image classification models that provide improved parameter efficiency and faster training speeds compared to previous models. Efficientnet_v2 leverages Neural Architecture Search (NAS) to optimize the model size and

**Table 1. Data list.**

|  | AMD | DME | Normal |
|---|---|---|---|
| people number | 15 | 15 | 15 |
| picture number | 723 | 1101 | 1407 |

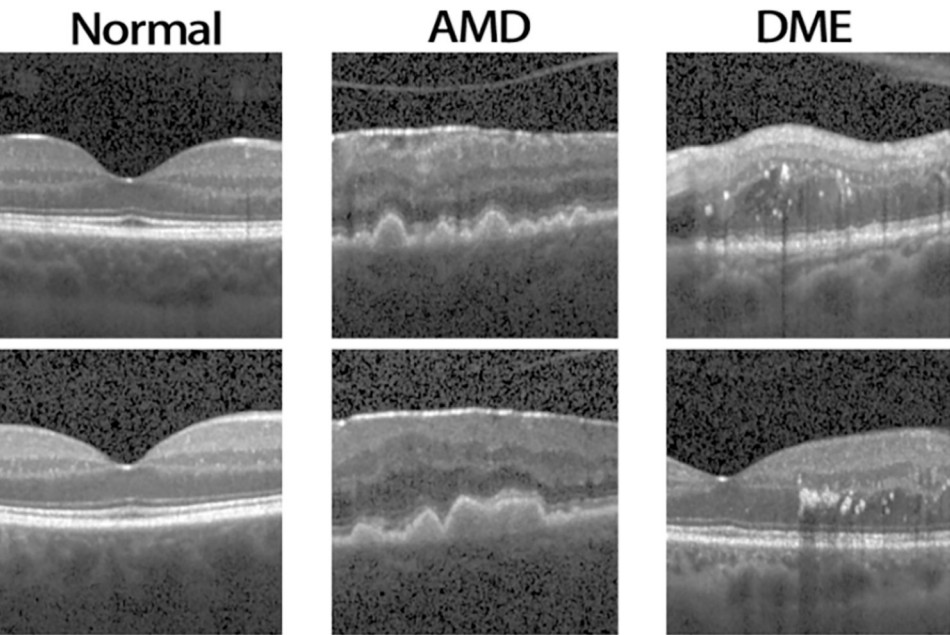

**Fig 2.** Examples of OCT images from the normal (column 1), AMD (column 2), and DME (column 3) datasets.

training speed, scaling to deliver swift training and inference. Table 2 summarizes the number of parameters for each of these three CNN models.

To explore the impact of migration learning on the prediction of fundus lesion types based on CNN networks, we compared the prediction performance of pre-trained models and re-trained models of the three frameworks mentioned above. On the one hand, we trained the three frameworks from scratch. To reduce the stress of training from scratch, we loaded the parameters of the pre-trained model trained from the ImageNet [42] database.

## The ensemble models

In this subsection, the performance of the ensemble model for predicting fundus lesions from OCT images is observed by 5-fold cross-validation. An ensemble model is an algorithm that combines the prediction results of two or more CNN models. The model proposed in this paper consists of a basic module and a prediction module. The basic module consists of three mutually independent CNN networks, namely Alexnet, Efficient_V2, and Resnet34. Each network is loaded with pre-trained weights, all use the same training set data and test set data, and all have hyperparameters of 100 epochs, 32 batch sizes, and 224*224 input channel sizes. The prediction results of the three networks (see Eq 1) are fed into the prediction module as the output of the basic model, which is mathematically represented as following:

$$O^i_{m_j} = (p^i_{1m_j}, p^i_{2m_j}, \cdots, p^i_{km_j}) \tag{1}$$

**Table 2. Parameters of the studied CNN models.**

| Model | Parameters (Millions) | Input image size |
|---|---|---|
| Alexnet | 61.1 | 3 x 224 x 224 |
| Efficientnet_v2 | 20.2 | 3 x 224 x 224 |
| Resnet34 | 21.3 | 3 x 224 x 224 |

Where the letter i stands for the $i$th sample. Model $m_j$ predicts the probability that the sample belongs to category k, denoted as $p^i_{km_j}$, $j \in \{1, \cdots, N\}$.

The prediction module uses a soft voting scheme to predict the class of the samples, namely calculating the average of the inputs as the final result, as shown in Eq 2.

$$O^i_{Ensemble} = \left[ \frac{\sum_{j=1}^{N} p^i_{1m_j}}{N}, \frac{\sum_{j=1}^{N} p^i_{2m_j}}{N}, \cdots \frac{\sum_{j=1}^{N} p^i_{km_j}}{N} \right] \tag{2}$$

Where $O^i_{Ensemble}$ denotes the probability that the sample belongs to each category. $O^i_{Ensemble}$ passes through softmax layer, and has the following equation:

$$p^i_{j,Ensemble} = \frac{O^i_{j,Ensemble}}{\sum_{j=1}^{3} O^i_{j,Ensemble}} \tag{3}$$

The catergorical loss function is used to calculate the loss between the ground truth and the predicted label. This cross entropy loss function is depicted as follows:

$$Loss = -\frac{1}{N_{data}} \sum_{i=1}^{N_{data}} \sum_{c=1}^{3} (y^T_{i,c} \log(p^i_{c,Ensemble})) \tag{4}$$

To increase the interpretability and intuitiveness of CNN network predictions, two popular visualization methods are used. One such technique is the CAM [43], which displays the contribution distribution of model output through a heat map. The heat map uses color to represent the contribution, where red represents a large contribution and blue represents a small contribution. However, CAM relies on a global average ensemble layer [44], which is a drawback. Another technique called Grad-CAM [45] overcomes this limitation by not requiring changes to the existing model structure, making it applicable to any CNN-based architecture. We use both methods to visualize feature maps, with Grad-CAM helping to understand what the model is learning. Unlike CAM, Grad-CAM is a class-specific localization technique that uses gradient information from the last convolutional layer of the CNN to understand the interest decisions of each neuron. The Grad-CAM visualization can be mathematically described as follows:

$$L = ReLU\left(\sum_k \alpha^{c_m}_k A^k_m\right) \tag{5}$$

$$\alpha^{c_m}_k = \frac{1}{z} \sum_i \sum_j \frac{\partial y^{c_m}}{\partial A^k_{m,ij}} \tag{6}$$

The weight $\alpha^{c_m}_k$ represents a partial linearization of the deep network downstream from $A^k_m$, and capture the importance of featrue map k from $c_m$ target class of model m.

## Experimental settings

The K-fold cross validation is used to give a more accurate measurement for model performance. The entire data set available in this assessment methodology is divided into K-sub parts during the training itself (K = 1,2,3,...). Each sub-section will then be viewed as a validation set for each iteration. The K value used in this work is 5. The performance of the experiments in this study was evaluated by accuracy, precision, recall, and F1 score [46]. Their

mathematical equations are given as following:

$$Accuracy = \frac{TP + TN + FP + FN}{TP + TN} \times 100 \tag{7}$$

$$Precision = \frac{TP}{TP + FP} \times 100 \tag{8}$$

$$Recall = \frac{TP}{TP + FN} \times 100 \tag{9}$$

$$F1\_score = \frac{TP + TN + FP + FN}{TP + TN} \times 100 \tag{10}$$

where True positive (TP) and true negative (TN) indicate accurately classified records, while false positive (FP) and false negative (FN) indicates information that has been wrongly classified. All simulations are carried out on a Nvidia 3060 GPU with 12 GB memory.

## Results

In this section, we explain the set of experiments conducted to evaluate our method and also validated its overall robustness and adaptability.

### The pre-training improves the baseline

To more comprehensively compare the impact of pre-training on model performance, we used three different CNN networks, Alexnet, Efficientnet_v2, and Resnet34. Figs 3–5 illustrate the performance comparison of three CNN networks between with and without pretrain in each class. One can see that the performance of all classes is significantly improved after loading pre-training parameters. In contrast, DME class has the best performance among them. The overall performance of all three networks is calculated and demonstrated in Tables 3 and 4. It is observed that the performance of all three networks was improved after loading pre-training parameters. Among them, the Resnet34 network loaded with pre-training has the strongest recognition ability, with an average AUC of 99.6%. The Efficientnet_v2 network has the most obvious improvement, with an AUC increase of 9.22% and a verification rate increase from 74.51% to 95.37%. After loading pre-training, the standard deviation (SD) of Alexnet, Efficientnet_v2, and Resnet34 models decreased by 3.45%, 2.97%, and 2.1%, respectively. Fig 6 makes the accuracy comparison of three CNN models. It also demonstrates the Resnet34 network loaded with pre-training has the strongest recognition ability. This suggests that pre-training has improved the robustness of these models.

### Analysis of the effectiveness of ensemble models based on pre-training methods

To evaluate the predictive performance of the ensemble model more completely, we compared it with the individual networks both overall and in a single category. Table 5 summarizes the performance of the three CNN models alone and the ensemble model. We observed that the ensemble model obtained the best classification performance with precision: 97.9±1.89, recall: 97.89±1.89. 97.89±1.89, F1_score: 97.88±1.9, AUC: 99.82±0.21. The ensemble model compared Alexnet, Efficientnet_v2 and Resnet34, and the F1 score has a significant improvement of 2.38, 2.55 and 1.17. Fig 7 shows the accuracy of the three CNNs models and the ensemble model. We observe that the ensemble model not only improves the accuracy, but also further

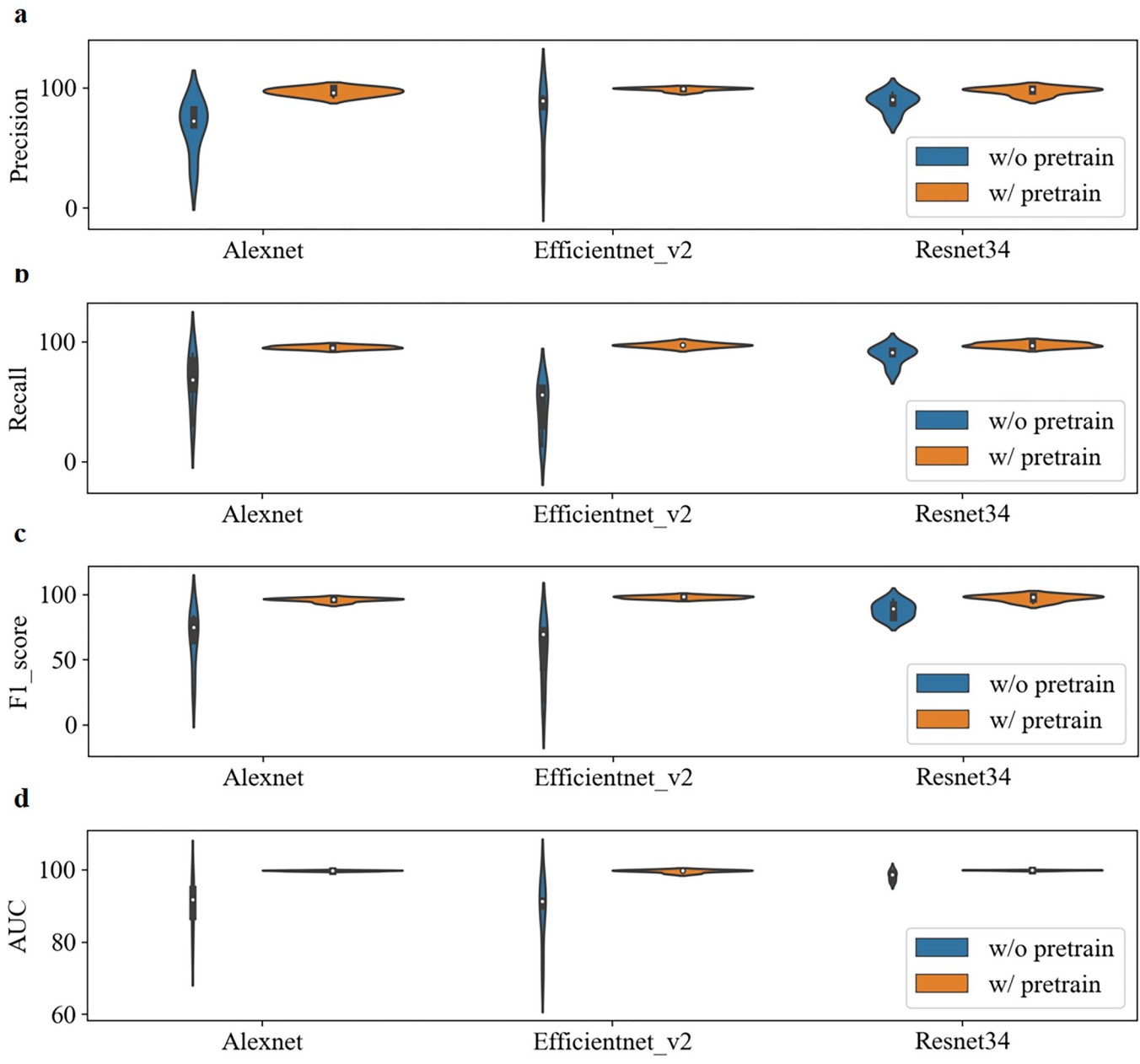

**Fig 3. Performance comparison of three CNN networks between with and without pretrain in AMD class.**

improves the robustness of the model. Comparing Alexnet, Efficientnet_v2, and Resnet34, it improves the classification accuracy by 2.38%, 2.52%, and 1.15%, and reduces the standard deviation by 0.47%, 0.91%, and 0.15%. Table 6 shows the performance of the three CNN models and the ensemble model in the three categories, AMD, DME and normal. We can see that the ensemble model improves the classification performance of each category. Especially for the normal category, the F1 score of the ensemble model is 98.57%, which is the maximum among all experiments. The ensemble model demonstrated the greatest improvement in recognizing DME, achieving an increase in F1 score of 3.55%, 3.69%, and 1.86% compared to Alexnet, Efficientnet_v2, and Resnet34, respectively.

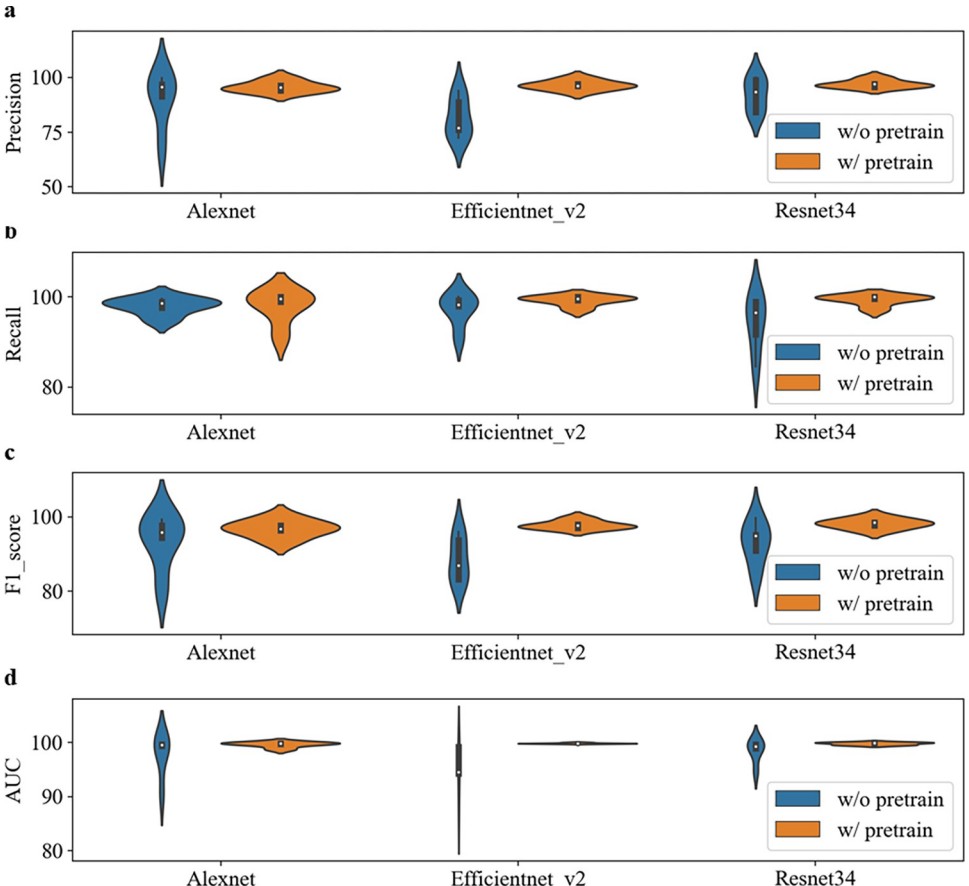

**Fig 4. Performance comparison of three CNN networks between with and without pretrain in Normal class.**

In addition to comparing with existing Alexnet, Efficientnet_v2, and Resnet34 CNN models, we further compared it with other existing methods. In [47], Wang et al. used N-Gram-Based Model for OCT classification, and the optimal accuracy obtained was 93.3%, and the AUC was 99.85%. In [48], the authors applied the Surrogate-Assisted CNN model for OCT classification, and obtained an accuracy of 95.09% and an AUC value of 98.56%. The optimal accuracy of the algorithm recommended in this article is greater than 98%, and the AUC is greater than 98.8%. It is significantly better than the above two models. This fully demonstrates the effectiveness of the proposed ensemble learning method.

## Model visualization

To increase the interpretability of the results even more, we visualized them using CAM and Grad-CAM, examples of which are shown in Fig 8. Where (a) to (c) are AMD, DME, and Noramal, the first column is the original image, the second column is the heat map of CAM, and the third column is the heat map of Grad-CAM. The highlighted areas in the heat map are shown in red and the weak areas are shown in blue. We observe that CAM is more blurred, while Grad-CAM shows finer information and more accurate details about the location and boundaries of the target lesion.

Grad-CAM generates heatmap-like results, highlighting the image areas where the model activates a specific category, offering insights into spatial locations. However, Grad-CAM

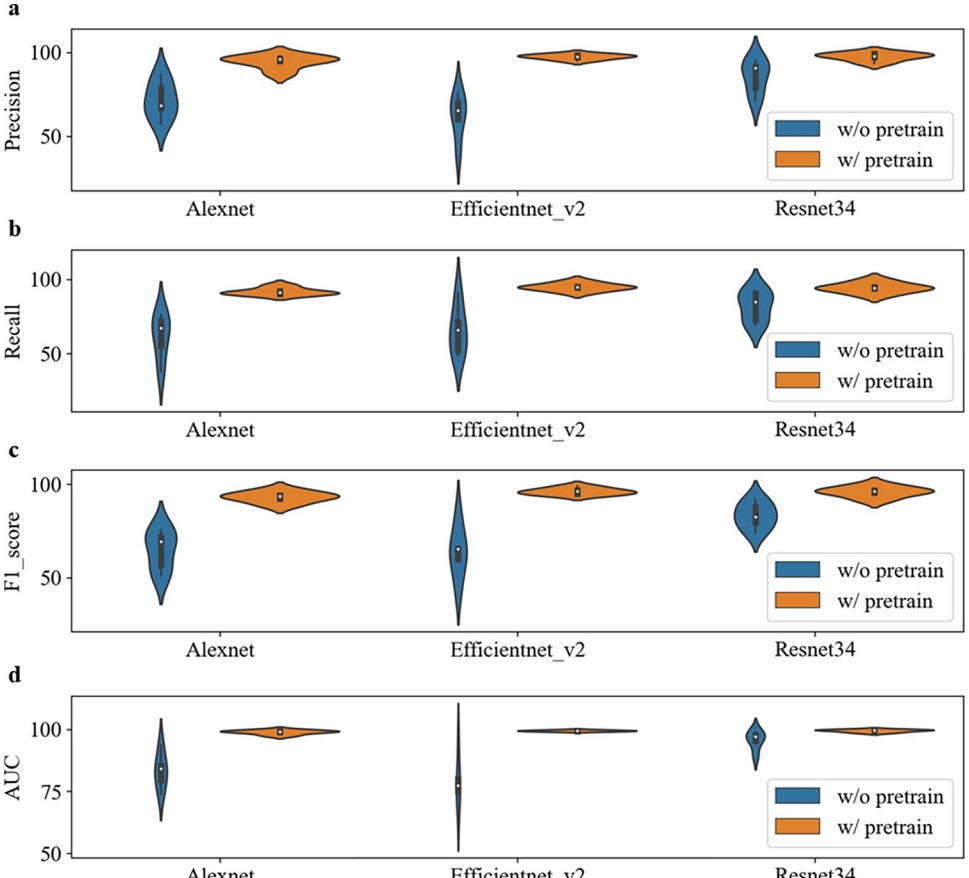

**Fig 5. Performance comparison of three CNN networks between with and without pretrain in DME class.**

results can sometimes appear coarse, complicating the identification of precise features guiding the model's decisions. In contrast, LIME [49] specializes in interpreting individual predictions, offering explanations that pinpoint pixel regions influencing each prediction. The two approaches complement each other: LIME delivers detailed, granular explanations, while Grad-CAM provides intuitive spatial information. Together, they may create a more comprehensive understanding of the model's decision-making process.

## Discussion

This paper describes a new method for identifying macular lesion types based on OCT images. The method successfully identifies cases of AMD, DME and Normal. The proposed method does not rely on segmentation of the internal retinal layers, but utilizes an easy-to-implement ensemble classification method based on a pre-training approach.

**Table 3. Performance of three CNN networks without pretrain (Mean±SD).**

| Model | Precision | Recall | F1_score | AUC |
| --- | --- | --- | --- | --- |
| Alexnet | 79.17±8.11 | 78.69±7.83 | 77.98±8.44 | 92.02±3.89 |
| Efficientnet_v2 | 74.80±8.64 | 74.51±8.09 | 72.80±9.11 | 89.87±3.66 |
| Resnet34 | 89.35±5.63 | 88.80±5.75 | 88.75±5.78 | 97.35±2.49 |

**Table 4. Performance of three CNN networks with pretrain (Mean±SD).**

| Model | Precision | Recall | F1_score | AUC |
|---|---|---|---|---|
| Alexnet | 95.58±2.33 | 95.51±2.36 | 95.5±2.35 | 99.48±0.44 |
| Efficientnet_v2 | 95.53±2.67 | 95.37±2.8 | 95.33±2.84 | 99.09±0.69 |
| Resnet34 | 96.84±1.96 | 96.74±2.04 | 96.71±2.08 | 99.6±0.39 |

It is noted that the model is robust to the input and that exactly the same algorithm parameters are used in all experiments. As explained in Section 2.1, we did not perform image improvement operations such as denoising, and also cropped all input images to a fixed image size. However, our algorithm still achieves perfect sensitivity and high specificity

In Section 2.2, to create the ensemble-based architecture, we used a basic module consisting of three CNN models and a prediction module. Experimental results on the same dataset show that the ensemble network improves the classification accuracy compared to a standalone

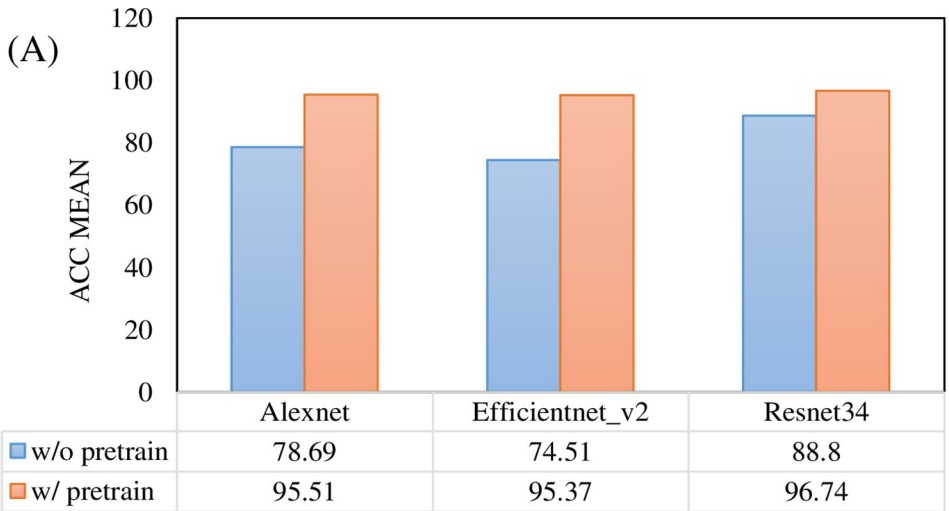

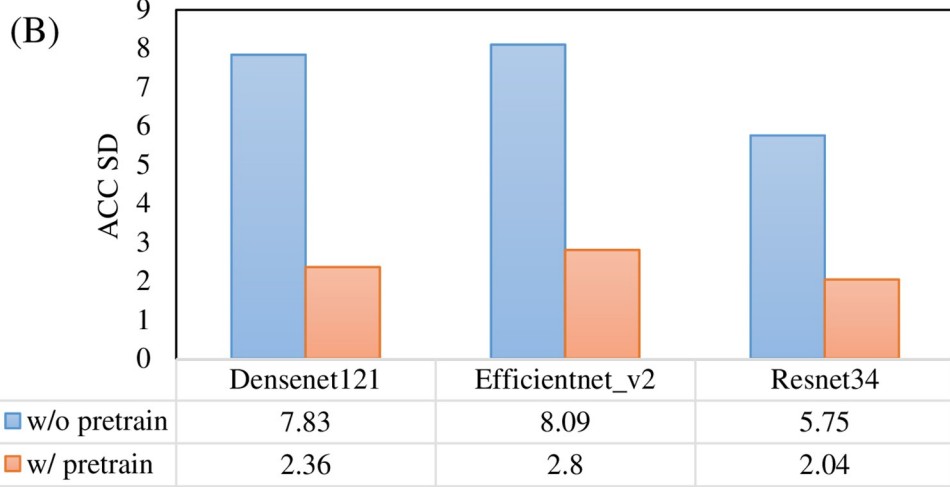

**Fig 6. Accuracy comparison of three CNN models between with and without pretrain.** (A) Mean value. (B) Standard deviance value.

**Table 5. Performance of three CNN models and ensemble model with pretraining (Mean±SD).**

| Model | Precision | Recall | F1_score | AUC |
|---|---|---|---|---|
| Alexnet [38] | 95.58±2.33 | 95.51±2.36 | 95.5±2.35 | 99.48±0.44 |
| Efficientnet_v2 [34] | 95.53±2.67 | 95.37±2.8 | 95.33±2.84 | 99.09±0.69 |
| Resnet34 [41] | 96.84±1.96 | 96.74±2.04 | 96.71±2.08 | 99.6±0.39 |
| Ensemble | 97.9±1.89 | 97.89±1.89 | 97.88±1.9 | 99.82±0.21 |

single CNN model. In addition, the method avoids the use of a large number of example images and more convergence by loading network parameters pre-trained on the ImageNet dataset, and achieves comparable performance.

The visualization of Grad-CAM also shows the ability of the model to learn different fundus lesion types. Due to the ensemble approach of the proposed model, the training and testing time is longer compared to a stand-alone single model. In addition, the memory requirements for training the model are high. This is a drawback of the model.

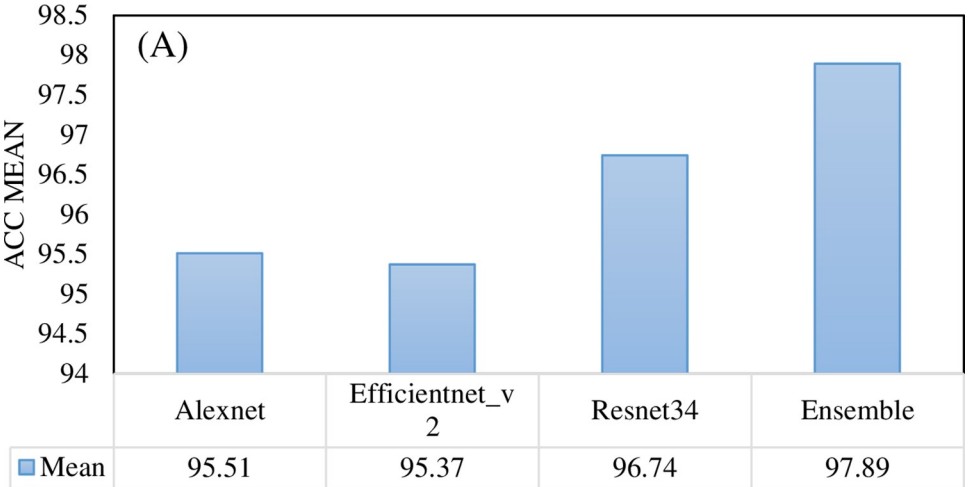

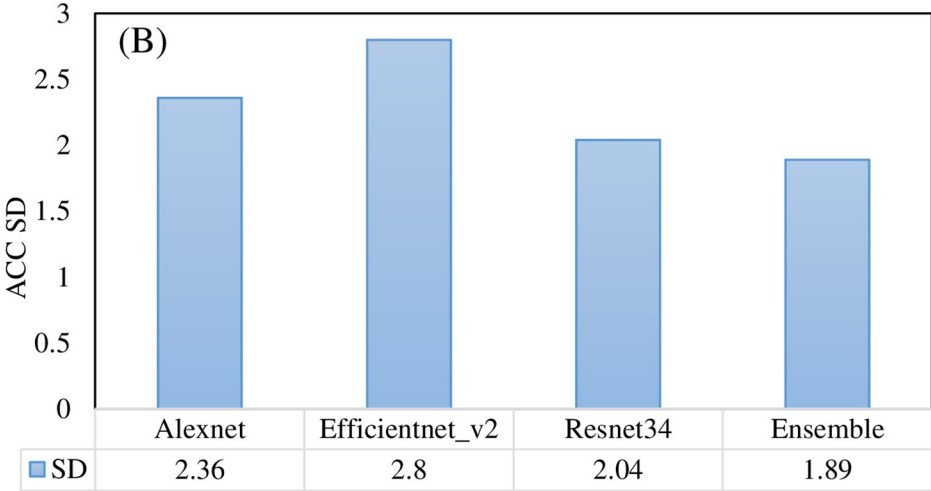

**Fig 7. Accuracy performance of three CNNs models and ensemble model.** (A) Mean value. (B) Standard deviance value.

**Table 6. Performance of three CNNs models and ensemble model with pretraining to each class (Mean±SD).**

| Model | Precision | Recall | F1_score | AUC |
|---|---|---|---|---|
| (a) AMD class | | | | |
| Alexnet | 96.8±3.31 | 95.66±1.42 | 96.19±1.47 | 99.81±0.13 |
| Efficientnet_v2 | 95.75±4.81 | 94.35±6.55 | 94.82±3.05 | 98.96±1.41 |
| Resnet34 | 96.94±4.06 | 98.26±1.62 | 97.57±2.49 | 99.94±0.09 |
| Ensemble | 98.32±2.49 | 97.87±1.34 | 98.07±1.19 | 99.97±0.04 |
| (b) DME class | | | | |
| Alexnet | 94.65±3.98 | 91.97±2.48 | 93.27±2.93 | 98.96±0.85 |
| Efficientnet_v2 | 94.56±4.89 | 92±5.73 | 93.13±3.84 | 98.81±0.86 |
| Resnet34 | 98.25±1.61 | 91.95±4.23 | 94.96±2.74 | 99.43±0.45 |
| Ensemble | 97.48±2.05 | 96.19±3.2 | 96.82±2.56 | 99.64±0.38 |
| (c) Normal class | | | | |
| Alexnet | 95.69±2.55 | 98.01±3.71 | 96.79±2.35 | 99.63±0.5 |
| Efficientnet_v2 | 95.91±2.97 | 98.69±1.53 | 97.27±2.07 | 99.57±0.48 |
| Resnet34 | 95.57±2.54 | 99.66±0.49 | 97.56±1.45 | 99.76±0.18 |
| Ensemble | 97.97±1.81 | 99.18±1.66 | 98.57±1.68 | 99.85±0.19 |

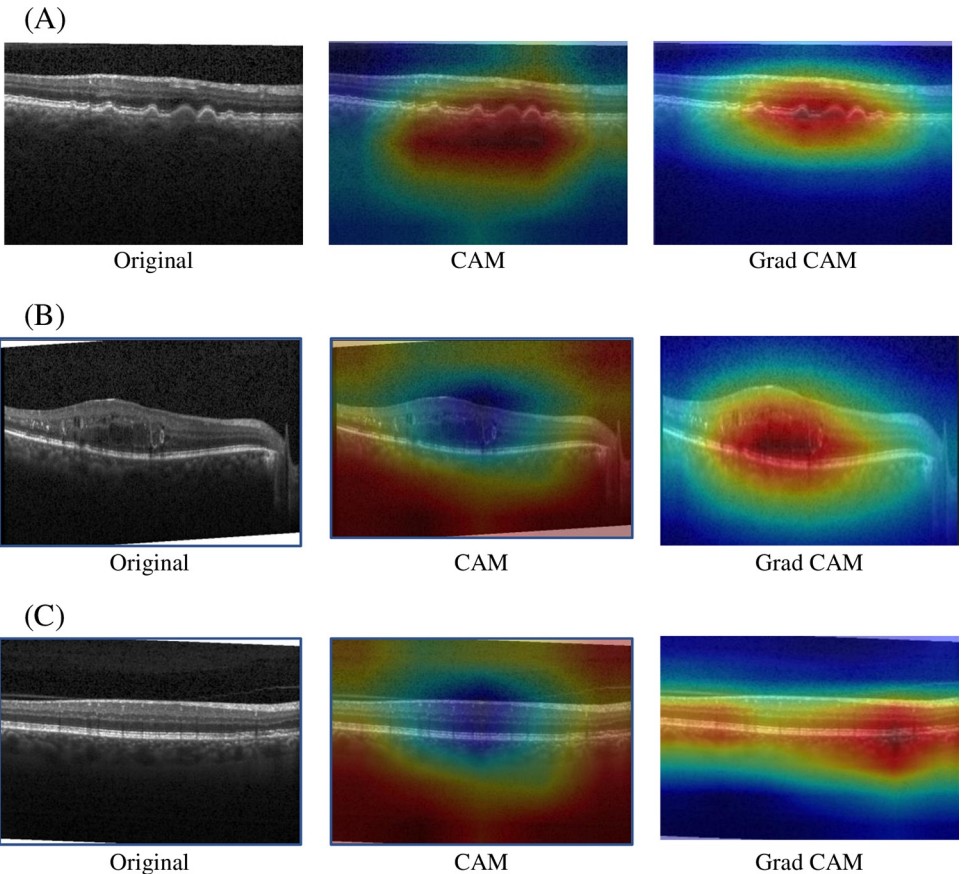

**Fig 8. Performance comparison between CAM and Grad-CAM. (A)** AMD OCT image. **(B)** DME OCT image. (C) Normal OCT image.

Our proposed approach exhibits promising results in the detection of retinal OCT images. However, it is crucial to acknowledge certain limitations. In this study, our primary focus was on detecting fundus diseases, specifically dry AMD and DME. Nonetheless, it is essential to consider the various other retinal diseases that may emerge within a clinical setting. The performance of the model and its applicability to other diseases still remain uncertain.

In our future work, we intend to enhance the dataset in order to improve generalization and encompass a broader spectrum of retinal diseases. Additionally, our aim is to perform practical validation in a clinical setting by seamlessly integrating artificial intelligence models into existing clinical workflows and effectively utilizing them. This will facilitate the incorporation of deep learning models into routine clinical practice.

In addition, in exploring the future trajectory for ensemble CNN models, it is evident that their potential extends across various domains and problem landscapes. One promising avenue for advancement lies in tailoring ensemble architectures to specific problem rather than pursuing a one-size-fits-all domains approach. This necessitates a shift towards domain-centric modifications wherein ensemble structures are customized to capture nuanced features inherent to the problem at hand. Moreover, challenges persist in enhancing the diversity and robustness of ensemble constituents while ensuring computational efficiency. To address these challenges, future Endeavors could focus on designing novel strategies to enhance diversity among individual models within ensembles and optimize their integration for improved predictive performance.

## Conclusions

In this paper, a new method is developed to accurately detect retinal diseases using deep migration learning based on an integrated network. The proposed method outperforms the results obtained using a single network in diagnosing retinal OCT images. In addition, the proposed method is fully automated. It can assist ophthalmologists in making diagnostic decisions. Future work in this study is to translate the proposed method into software that can be used by specialists in medical centers as a screening tool and to provide second opinions. Finally, we expect that the proposed approach could be potentially applied to medical image classifications, such as chest X-rays and MRIs, to assist clinicians in making diagnostic decisions.

## Author Contributions

**Conceptualization:** Jiasheng Yang, Xu Xiao.

**Data curation:** Jiasheng Yang.

**Funding acquisition:** Jiasheng Yang, Geng Tian.

**Investigation:** Jiasheng Yang, Guanfang Wang, Xu Xiao.

**Methodology:** Jiasheng Yang, Xu Xiao, Geng Tian.

**Project administration:** Meihua Bao, Geng Tian.

**Resources:** Meihua Bao.

**Software:** Jiasheng Yang.

**Supervision:** Jiasheng Yang, Geng Tian.

**Validation:** Guanfang Wang.

**Visualization:** Jiasheng Yang, Xu Xiao.

**Writing – original draft:** Jiasheng Yang, Guanfang Wang.

**Writing – review & editing:** Jiasheng Yang, Guanfang Wang, Xu Xiao, Meihua Bao, Geng Tian.

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
