## [Decision Letter · Decision Letter 0]

26 Apr 2023

PONE-D-23-05848Explainable ensemble learning method for OCT detection with transfer learningPLOS ONE

Dear Dr. yang,

Thank you for submitting your manuscript to PLOS ONE. After careful consideration, we feel that it has merit but does not fully meet PLOS ONE’s publication criteria as it currently stands. Therefore, we invite you to submit a revised version of the manuscript that addresses the points raised during the review process.

We look forward to receiving your revised manuscript.

Kind regards,

Ali Mohammad Alqudah

Academic Editor

PLOS ONE

Journal Requirements:

"This research is supported by AHUT research fund(DT2200000873)."

3. Please expand the acronym “AHUT” (as indicated in your financial disclosure) so that it states the name of your funders in full.

"This research is supported by AHUT research fund(DT2200000873)."

"This research is supported by AHUT research fund(DT2200000873)."

'I have read the journal's policy and the authors of this manuscript have the following competing interests: The authors declare that they have no known competing financial interests or personal 

relationships that could have appeared to influence the work reported in this paper."

7. PLOS requires an ORCID iD for the corresponding author in Editorial Manager on papers submitted after December 6th, 2016. Please ensure that you have an ORCID iD and that it is validated in Editorial Manager. To do this, go to ‘Update my Information’ (in the upper left-hand corner of the main menu), and click on the Fetch/Validate link next to the ORCID field. This will take you to the ORCID site and allow you to create a new iD or authenticate a pre-existing iD in Editorial Manager. Please see the following video for instructions on linking an ORCID iD to your Editorial Manager account: https://www.youtube.com/watch?v=_xcclfuvtxQ

Reviewers' comments:

Reviewer's Responses to Questions

**Comments to the Author**

1. Is the manuscript technically sound, and do the data support the conclusions?

Reviewer #1: Yes

Reviewer #2: Yes

2. Has the statistical analysis been performed appropriately and rigorously? 

Reviewer #1: Yes

Reviewer #2: Yes

3. Have the authors made all data underlying the findings in their manuscript fully available?

Reviewer #1: Yes

Reviewer #2: Yes

4. Is the manuscript presented in an intelligible fashion and written in standard English?

Reviewer #1: Yes

Reviewer #2: No

5. Review Comments to the Author

Reviewer #1: The authors have done good work on the title “Explainable ensemble learning method for OCT detection with transfer learning”. It will add new knowledge and new areas of research to the subject area compared with other published material.

However, i have some minor concerns:

1. It would be more appropriate for the authors to define abbreviations upon first appearance in the main text such as degeneration (AMD), diabetic macular edema (DME), Optical coherence tomography (OCT) and convolutional neural network (CNN).

2. Figure 1 need more enhancement to be of good quality; image resizing.

3. It would be more appropriate for the authors to standardize the writing of the CNN networks’ names “Efficient_V2, and Resnet34” across the whole manuscript.

4. The authors have inserted Equations 1,3, 4, 5 and 6 in the section of “The Ensemble models”, but it was not cited inside the manuscript. Kindly check it and perform the required amendment.

5. The authors have inserted Figures 3 and 4 in the section of “The Ensemble models”, but it was not cited inside the manuscript. Kindly check the guideline of the PLOS ONE journal and perform the required amendment.

6. In the section of “Analysis of the effectiveness of Ensemble models based on pre-training methods”, the authors stated that the ensemble model obtained the best classification performance with accuracy: 97.9±1.89. However, in Table 5, the authors represented the precision value not the accuracy value. Kindly check it and perform the required amendment.

7. In the section of “Analysis of the effectiveness of Ensemble models based on pre-training methods”, some of the calculations in the following sentences are not correct. Kindly check it and perform the required amendment.

“Efficientnet_v2 and Resnet34, and the F1 score has a significant improvement of 2.38, 2.55 and 1.15. Figure 5 shows the accuracy of the three CNNs models and the ensemble model. We observe that the ensemble model not only improves the accuracy, but also further improves the robustness of the model. Comparing Alexnet, Efficientnet_v2, and Resnet34, it improves the classification accuracy by 2.37%, 2.52%, and 1.15%,”.

8. Authors should include the discussion section to interpret of their results and explain how the results relate to the hypothesis presented as the basis of the study and provide a succinct explanation of the implications of the findings, particularly in relation to previous related studies and potential future directions for research. I encourage the authors to present and discuss their findings concisely in discussion section.

9. Moderate editing is required throughout the manuscript, for example:

1. In the abstract, “The accuracy and interpretability of artificial intelligence are fundamental to revolutionizing Optical Coherence Tomography (OCT) image detection, significantly reducing the grueling manual labor required of clinicians…….., with 15 samples in each category”. Moderate grammar editing is required.

2. “AMD) and DME are common a”. Moderate editing is required.

3. “For this study, the proposed algorithm was designed and evaluated on a publicly 138 available dataset[29].” Moderate editing is required.

4. “Specifically, the image preprocessing process in this paper is as follows. First, data enhancement is performed on the image by horizontal flip, vertical flip and a….). Moderate editing is required.

5. “Alexnet[30-32] is comprised of eight layers, including five convolutional layers and three fully connected layers”. Moderate editing is required.

6. “EfficientNetV2[34] is a family of image classification models, which offer superior parameter efficiency and faster…”. Moderate editing is required.

7. “On the one hand, we trained the three frameworks from scratch. On the other hand, we loaded pre-trained model parameters to reduce the stress of training from scratch. The pre-training used in this paper was all performed on the ImageNet[35] database”. Moderate editing is required.

8. “The performances of our experiments are evaluated through Accuracy , Precision,”. Moderate editing is required.

9. “And the standard deviation (SD) of Alexnet, Efficient_v2 and Resnet34 models decreased by 3.45%, 254 2.97% and 2.1% after loading pre-training, which indicates the improved robustness 255 of these models”. Moderate editing is required.

10. “And the ensemble model has the most improvement inrecognizing DME, with an F1 score increase of 3.55%, 3.69% and 1.86% compared toAlexnet, Efficientnet_v2 and Resnet34”. Moderate editing is required.

Best regards,

Dr. Mai Abdel Haleem Abusalah

Faculty of Medical Allied Science,

Zarqa University,

Zarqa, 13110, Jordan.

Tel: +962-796862347

e-mail: ellamomo88@yahoo.com

Reviewer #2: This manuscript is an excellent piece of work that applies a cutting-edge integrated deep-learning model for categorizing eye conditions. It emphasizes how combining networks can increase prediction accuracy when compared to using individual networks alone and offers model learning presentation through heat mappings. Though the study conclusions are sound, the quality of the dataset used in this study may influence the results.

Major issues:

1. The publicly accessible dataset used in this work appears to come from Spectral Domain OCT, which is particularly susceptible to mirror artifacts and speckle noise. It is encouraged to specify the inclusion or exclusion of such artifacts in the pre-possessing section as they may have an influence on the prediction accuracy.

2. The study fails to address how the findings relate to previous research in this area. The authors can rewrite their Introduction and Discussion to include recent references from the related topic.

Minor issues:

1. Line 109: vitreous warts; Can be reworded

2. Line 115: age-related fundus macular degeneration; Can be reworded

3. Line 148: Phrasing issue

4. Line 151: Table 1: people number, picture number; Can be reworded

5. Lines 173 and 174: Can be condensed into a paragraph (Incomplete comparison)

6. Line 178: Table 2: Units for the input image size (Dimensions/pixels)

7. Line 322 and 336: Typo for ‘normal’

8. Line 340: phrasing issue

9. Line 343: Perfect sensitivity; Can be reworded

6. PLOS authors have the option to publish the peer review history of their article (what does this mean?). If published, this will include your full peer review and any attached files.

Reviewer #1: **Yes: **MAI ABDEL HALEEM ABUSALAH

Reviewer #2: No

---

## [Author Response · Author response to Decision Letter 0]

7 Jun 2023

Reviewer#1

The authors have done good work on the title “Explainable ensemble learning method for OCT detection with transfer learning”. It will add new knowledge and new areas of research to the subject area compared with other published material. 

1.It would be more appropriate for the authors to define abbreviations upon first appearance in the main text such as degeneration (AMD), diabetic macular edema (DME), Optical coherence tomography (OCT) and convolutional neural network (CNN).

Response: Thank you for your comment. We have defined the abbreviations that appear for the first time in the main text.

2.Figure 1 need more enhancement to be of good quality; image resizing.

Response: Thanks for pointing out this! We have improved the quality of figure 1.

3.It would be more appropriate for the authors to standardize the writing of the CNN networks’ names “Efficient_V2, and Resnet34” across the whole manuscript.

Response: Thank you for the suggestion!We have standardized the writing of the CNN network name "Efficientnet_v2, and Resnet34" throughout the manuscript.

4.The authors have inserted Equations 1,3, 4, 5 and 6 in the section of “The Ensemble models”, but it was not cited inside the manuscript. Kindly check it and perform the required amendment.

Response: Thanks a lot! We have checked and made amendement.

5.The authors have inserted Figures 3 and 4 in the section of “The Ensemble models”, but it was not cited inside the manuscript. Kindly check the guideline of the PLOS ONE journal and perform the required amendment.

Response: Thank you for your comment. We have made amendement. It is demonstrated as following:

“To more comprehensively compare the impact of pre-training on model performance, we used three different CNN networks, Alexnet, Efficientnet_v2, and Resnet34. Fig. 3 illustrates the performance comparison of three CNN networks between with and without pretrain in each class. One can see that the performance of all classes is significantly improved after loading pre-training parameters. In contrast, DME class has the best performance among them. The overall performance of all three networks is calculated and demonstrated in Tables 3 and 4. It is observed that the performance of all three networks was improved after loading pre-training parameters. Among them, the Resnet34 network loaded with pre-training has the strongest recognition ability, with an average AUC of 99.6%. The Efficientnet_v2 network has the most obvious improvement, with an AUC increase of 9.22% and a verification rate increase from 74.51% to 95.37%. After loading pre-training, the standard deviation (SD) of Alexnet, Efficientnet_v2, and Resnet34 models decreased by 3.45%, 2.97%, and 2.1%, respectively. Fig. 4 makes the accuracy comparison of three CNN models. It also demonstrates the Resnet34 network loaded with pre-training has the strongest recognition ability. This suggests that pre-training has improved the robustness of these models.”

6.In the section of “Analysis of the effectiveness of Ensemble models based on pre-training methods”, the authors stated that the ensemble model obtained the best classification performance with accuracy: 97.9±1.89. However, in Table 5, the authors represented the precision value not the accuracy value. Kindly check it and perform the required amendment.

Response: Thank you for your comment. It is a typo error. We rewrite the word “accuracy” to be “precision” in the updated manuscript.

7.In the section of “Analysis of the effectiveness of Ensemble models based on pre-training methods”, some of the calculations in the following sentences are not correct. Kindly check it and perform the required amendment.

“Efficientnet_v2 and Resnet34, and the F1 score has a significant improvement of 2.38, 2.55 and 1.15. Figure 5 shows the accuracy of the three CNNs models and the ensemble model. We observe that the ensemble model not only improves the accuracy, but also further improves the robustness of the model. Comparing Alexnet, Efficientnet_v2, and Resnet34, it improves the classification accuracy by 2.37%, 2.52%, and 1.15%,”.

Response: Thank you for the suggestion! We double check and rewrite it as following:

“The ensemble model compared Alexnet, Efficientnet_v2 and Resnet34, and the F1 score has a significant improvement of 2.38, 2.55 and 1.17. ”.

8.Authors should include the discussion section to interpret of their results and explain how the results relate to the hypothesis presented as the basis of the study and provide a succinct explanation of the implications of the findings, particularly in relation to previous related studies and potential future directions for research. I encourage the authors to present and discuss their findings concisely in discussion section.

Response: Thank you very much for pointing out the problem! We have discussion section to demonstrate our findings.

Reviewer#2

Moderate editing is required throughout the manuscript, for example: 

1.In the abstract, “The accuracy and interpretability of artificial intelligence are fundamental to revolutionizing Optical Coherence Tomography (OCT) image detection, significantly reducing the grueling manual labor required of clinicians…….., with 15 samples in each category”. Moderate grammar editing is required. 

Response: Thank you for the suggestion! We have rewritten these sentences as follows:

The accuracy and interpretability of artificial intelligence (AI) are crucial for the advancement of optical coherence tomography (OCT) image detection, as it can greatly reduce the manual labor required by clinicians. By prioritizing these aspects during development and application, we can make significant progress towards streamlining the clinical workflow. In this paper, we propose an explainable ensemble approach that utilizes transfer learning to detect fundus lesion diseases through OCT imaging. Our study utilized a publicly available OCT dataset consisting of normal subjects, patients with dry age-related macular degeneration (AMD), and patients with diabetic macular edema (DME), each with 15 samples.

2.“AMD) and DME are common a”. Moderate editing is required. 

Response: Thank you for your comment. We have rewritten these sentences as follows:

Age-related macular degeneration (AMD) and diabetic macular edema (DME) are prevalent eye conditions that are becoming more common among elderly individuals and those with diabetes globally.

3.“For this study, the proposed algorithm was designed and evaluated on a publicly 138 available dataset[29].” Moderate editing is required. 

Response: Thank you for your comment. We have rewritten these sentences as follows:

In this study, we designed and evaluated the proposed algorithm using a publicly available dataset [29].

4.“Specifically, the image preprocessing process in this paper is as follows. First, data enhancement is performed on the image by horizontal flip, vertical flip and a….). Moderate editing is required.

Response: Thanks a lot! We have rewritten these sentences as follows:

In this paper, the image preprocessing process involves data enhancement through horizontal and vertical flip as well as rotation within a range of -15° to 15°. 

5.“Alexnet[30-32] is comprised of eight layers, including five convolutional layers and three fully connected layers”. Moderate editing is required.

Response: Thanks for pointing out this! We have rewritten these sentences as follows:

Alexnet [30-32] consists of a total of eight layers, which comprise of five convolutional layers and three fully connected layers. 

6.“EfficientNetV2[34] is a family of image classification models, which offer superior parameter efficiency and faster…”. Moderate editing is required.

Response: Thank you for the suggestion!We have rewritten these sentences as follows:

Efficientnet_v2 [34] is a family of image classification models that provide improved parameter efficiency and faster training speeds compared to previous models. 

7.“On the one hand, we trained the three frameworks from scratch. On the other hand, we loaded pre-trained model parameters to reduce the stress of training from scratch. The pre-training used in this paper was all performed on the ImageNet[35] database”. Moderate editing is required.

Response: Thank you for your comment.We have rewritten these sentences as follows:

To reduce the stress of training from scratch, we loaded the parameters of the pre-trained model trained from the ImageNet[35] database.

8.“The performances of our experiments are evaluated through Accuracy , Precision,”. Moderate editing is required.

Response: Thank you for your comment.We have rewritten these sentences as follows:

The performance of the experiments in this study was evaluated by accuracy, precision, recall, and F1 score.

9.“And the standard deviation (SD) of Alexnet, Efficient_v2 and Resnet34 models decreased by 3.45%, 254 2.97% and 2.1% after loading pre-training, which indicates the improved robustness 255 of these models”. Moderate editing is required.

Response: Thank you for your comment.We have rewritten these sentences as follows:

After loading pre-training, the standard deviation (SD) of Alexnet, Efficientnet_v2, and Resnet34 models decreased by 3.45%, 2.97%, and 2.1%, respectively. This suggests that pre-training has improved the robustness of these models.

10.“And the ensemble model has the most improvement inrecognizing DME, with an F1 score increase of 3.55%, 3.69% and 1.86% compared toAlexnet, Efficientnet_v2 and Resnet34”. Moderate editing is required.

Response: Thank you for your comment.We have rewritten these sentences as follows:

The ensemble model demonstrated the greatest improvement in recognizing DME, achieving an increase in F1 score of 3.55%, 3.69%, and 1.86% compared to Alexnet, Efficientnet_v2, and Resnet34, respectively.

---

## [Decision Letter · Decision Letter 1]

19 Oct 2023

PONE-D-23-05848R1Explainable ensemble learning method for OCT detection with transfer learningPLOS ONE

Dear Dr. yang,

Thank you for submitting your manuscript to PLOS ONE. After careful consideration, we feel that it has merit but does not fully meet PLOS ONE’s publication criteria as it currently stands. Therefore, we invite you to submit a revised version of the manuscript that addresses the points raised during the review process.

We look forward to receiving your revised manuscript.

Kind regards,

Ali Mohammad Alqudah

Academic Editor

PLOS ONE

Journal Requirements:

**Additional Editor Comments:**

Dear Authors,

Please read the reviewer comments and reply to them.

Thank you

Reviewers' comments:

Reviewer's Responses to Questions

**Comments to the Author**

1. If the authors have adequately addressed your comments raised in a previous round of review and you feel that this manuscript is now acceptable for publication, you may indicate that here to bypass the “Comments to the Author” section, enter your conflict of interest statement in the “Confidential to Editor” section, and submit your "Accept" recommendation.

Reviewer #1: All comments have been addressed

Reviewer #3: (No Response)

2. Is the manuscript technically sound, and do the data support the conclusions?

Reviewer #1: Yes

Reviewer #3: Yes

3. Has the statistical analysis been performed appropriately and rigorously? 

Reviewer #1: Yes

Reviewer #3: No

4. Have the authors made all data underlying the findings in their manuscript fully available?

Reviewer #1: Yes

Reviewer #3: Yes

5. Is the manuscript presented in an intelligible fashion and written in standard English?

Reviewer #1: Yes

Reviewer #3: Yes

6. Review Comments to the Author

Reviewer #1: The authors have done good work on the title “Explainable ensemble learning method for OCT detection with transfer learning”. It will add new knowledge and new areas of research to the subject area compared with other published material.

The authors have adequately addressed all comments and performed the required amendments; hence I highly recommend accepting this interesting article.

Best regards,

Dr. Mai Abdel Haleem Abusalah

Faculty of Medical Allied Science,

Zarqa University,

Zarqa, 13110, Jordan.

Tel: +962-796862347

e-mail: ellamomo88@yahoo.com

Best regards,

Reviewer #3: For Figure 6, please add the color code to the maps. Also, please add discussion about “why” as well as if this can be further enhanced using other explainable methods, e.g., LIME.

The discussion section lacks many aspects, such as limitation, how the methods can be applied to clinical settings, what is the future work, etc.

I am not sure why the authors add refs [43] and [44] to the conclusions section. Please make future direction as general to specific problem, not a specific approach, and explain more (in the discussion) what modifications should be done, challenges, etc.

Also, evaluation on a single data set is not suggested. The authors need to verify the robustness of their methods in additional data sets. There are publicly available OCTs data sets (e.g., UCSD dataset). Comparison with the peer methods that utilized the same data set should be included.

The authors need to separate the introduction from the related work. Additionally, the literature work lacks some approaches that employed ensemble/computational learning for OCT classification e.g., Huang et al. Front. Neurosci. 2023; Ai et al. Front. Neuroinform. 2022, Akinniyi et al., Bioengineering 2023; Kayadibi et al. 2023 Int. J. Comput. Intell. Syst .

Table 4, 5, and 6 need statistical analysis of the reported values and Figures 4&5 are not necessary to as Tables are sufficient.

Please review the manuscript writing and language. For example, instead of “L. R. Ashok et al.” it should be only “Ashok et al.”

The abbreviation should be doen one time, e.g., age-related macular degeneration (AMD) is used in page 3 and 4

7. PLOS authors have the option to publish the peer review history of their article (what does this mean?). If published, this will include your full peer review and any attached files.

Reviewer #1: **Yes: **MAI ABDEL HALEEM ABUSLAH

Reviewer #3: No

---

## [Author Response · Author response to Decision Letter 1]

4 Dec 2023

Dear Editor, Dear reviewers,

Thanks very much for taking your time to review this manu. I really appreciate all your comments and suggestions! Please find my itemized responses in below and my revisions/corrections in the re-submitted files.

Reviewer#3

(1) For Figure 6, please add the color code to the maps. Also, please add discussion about “why” as well as if this can be further enhanced using other explainable methods, e.g., LIME. 

Response: Thank you for expert advice! In the updated manuscript, we make the following supplements in the discussion: 

“Grad-CAM generates heatmap-like results, highlighting the image areas where the model activates a specific category, offering insights into spatial locations. However, Grad-CAM results can sometimes appear coarse, complicating the identification of precise features guiding the model's decisions. In contrast, LIME specializes in interpreting individual predictions, offering explanations that pinpoint pixel regions influencing each prediction. The two approaches complement each other: LIME delivers detailed, granular explanations, while Grad-CAM provides intuitive spatial information. Together, they may create a more comprehensive understanding of the model's decision-making process.”

(2) The discussion section lacks many aspects, such as limitation, how the methods can be applied to clinical settings, what is the future work, etc. 

Response: Thanks for pointing out this! According to this instruction, we make the following supplements in the discussion:

“Our proposed approach exhibits promising results in the detection of retinal OCT images. However, it is crucial to acknowledge certain limitations. In this study, our primary focus was on detecting fundus diseases, specifically dry AMD and DME. Nonetheless, it is essential to consider the various other retinal diseases that may emerge within a clinical setting. The performance of the model and its applicability to other diseases still remain uncertain.

In our future work, we intend to enhance the dataset in order to improve generalization and encompass a broader spectrum of retinal diseases. Additionally, our aim is to perform practical validation in a clinical setting by seamlessly integrating artificial intelligence models into existing clinical workflows and effectively utilizing them. This will facilitate the incorporation of deep learning models into routine clinical practice.”

(3)I am not sure why the authors add refs [43] and [44] to the conclusions section. Please make future direction as general to specific problem, not a specific approach, and explain more (in the discussion) what modifications should be done, challenges, etc. 

Response: Thanks for pointing out this! According to this instruction, we make the following supplements in the discussion:

“In exploring the future trajectory for ensemble CNN models, it is evident that their potential extends across various domains and problem landscapes. One promising avenue for advancement lies in tailoring ensemble architectures to specific problem rather than pursuing a one-size-fits-all domains approach . This necessitates a shift towards domain-centric modifications wherein ensemble structures are customized to capture nuanced features inherent to the problem at hand. Moreover, challenges persist in enhancing the diversity and robustness of ensemble constituents while ensuring computational efficiency. To address these challenges, future Endeavors could focus on designing novel strategies to enhance diversity among individual models within ensembles and optimize their integration for improved predictive performance.”

 (4) Also, evaluation on a single data set is not suggested. The authors need to verify the robustness of their methods in additional data sets. There are publicly available OCTs data sets (e.g., UCSD dataset). Comparison with the peer methods that utilized the same data set should be included. 

Response: Thanks for pointing out this! In the updated manuscript, we compare our method with other existing methods. At first, we compare the proposed method with Alexnet [38]，Efficientnet [34], Resnet34 [41] CNN models. Furthermore, we compare it with the other machine learning models . All demonstrate that the proposed ensemble methods has best performance. we make the following supplements in the introduction:

“In addition to comparing with existing Alexnet, Efficientnet_v2, and Resnet34 CNN models, we further compared it with other existing methods. In [47], Wang et al. used N-Gram-Based Model for OCT classification, and the optimal accuracy obtained was 93.3%, and the AUC was 99.85%. In [48], the authors applied the Surrogate-Assisted CNN model for OCT classification, and obtained an accuracy of 95.09% and an AUC value of 98.56%. The optimal accuracy of the algorithm recommended in this article is greater than 98%, and the AUC is greater than 98.8%. It is significantly better than the above two models. This fully demonstrates the effectiveness of the proposed ensemble learning method.”

 (5) The authors need to separate the introduction from the related work. Additionally, the literature work lacks some approaches that employed ensemble/computational learning for OCT classification e.g., Huang et al. Front. Neurosci. 2023; Ai et al. Front. Neuroinform. 2022, Akinniyi et al., Bioengineering 2023; Kayadibi et al. 2023 Int. J. Comput. Intell. Syst. 

Response: Thank you for your comment. According to the good instruction, we make the following supplements in the introduction:

Huang et al. [33] introduced a novel global attention block (GAB) that enhances classification performance when integrated with any CNN, resulting in a notable improvement of 3.7% accuracy compared with the EfficientNetV2B3 model. Ai et al. [34] proposed FN-OCT, a fusion network-based algorithm for retinal OCT classification that fuses predictions from InceptionV3, Inception-ResNet, and Xception classifiers with CBAM, employing three fusion strategies. On the dataset from UCSD, FN-OCT achieved 98.7% accuracy and 99.1% AUC, surpassing InceptionV3 by 5.3%. Akinniyi et al. [35] proposed a multi-stage classification network using OCT images for retinal image classification. Their architecture utilizes a pyramidal feature ensemble built on DenseNet for extracting multi-scale features. Evaluation on two datasets demonstrates advantages, achieving high accuracies for binary, three-class, and four-class classification. Kayadibi et al. [36] proposed a hybrid approach for retinal disease identification. It uses a fully dense fusion neural network (FD-CNN) and dual preprocessing to reduce speckle noise, extract features, and generate heat maps for diagnosis confidence.

(6) Table 4, 5, and 6 need statistical analysis of the reported values and Figures 4&5 are not necessary to as Tables are sufficient.

Response: Thank you for underlining this deficiency. Figure 4&5 illustrates Accuracy performance of three CNNs models and ensemble model, and Table 4, 5, and 6 illustrates Precision, Recall, F1_score and AUC performances. Accuracy and F1 score are both indicators used to evaluate the performance of classification models, but they focus on slightly different aspects. They are used together to provide a comprehensive evaluation of model performance. In Table 4, 5, and 6, we already did statical analysis including mean value and variance to demonstrate the effectiveness of the proposed ensemble model. 

(7) Please review the manuscript writing and language. For example, instead of “L. R. Ashok et al.” it should be only “Ashok et al.”

Response: Thanks a lot! We revised the manuscript in accordance with the reviewers' comments and carefully proof-read the manuscript.

(8) The abbreviation should be doen one time, e.g., age-related macular degeneration (AMD) is used in page 3 and 4.

Response: Thanks a lot! We have made correction according to the comments.

---

## [Decision Letter · Decision Letter 2]

8 Dec 2023

Explainable ensemble learning method for OCT detection with transfer learning

PONE-D-23-05848R2

Dear Dr. yang,

We’re pleased to inform you that your manuscript has been judged scientifically suitable for publication and will be formally accepted for publication once it meets all outstanding technical requirements.

Kind regards,

Ali Mohammad Alqudah

Academic Editor

PLOS ONE

Reviewers' comments:

Reviewer's Responses to Questions

**Comments to the Author**

1. If the authors have adequately addressed your comments raised in a previous round of review and you feel that this manuscript is now acceptable for publication, you may indicate that here to bypass the “Comments to the Author” section, enter your conflict of interest statement in the “Confidential to Editor” section, and submit your "Accept" recommendation.

Reviewer #3: All comments have been addressed

2. Is the manuscript technically sound, and do the data support the conclusions?

Reviewer #3: Yes

3. Has the statistical analysis been performed appropriately and rigorously? 

Reviewer #3: No

4. Have the authors made all data underlying the findings in their manuscript fully available?

Reviewer #3: Yes

5. Is the manuscript presented in an intelligible fashion and written in standard English?

Reviewer #3: Yes

6. Review Comments to the Author

Reviewer #3: No more comments. The authors provided rebuttal address the previous comments and the revised version has been updated accordingly

7. PLOS authors have the option to publish the peer review history of their article (what does this mean?). If published, this will include your full peer review and any attached files.

Reviewer #3: No

---

## [Editor Report · Acceptance letter]

4 Jan 2024

PONE-D-23-05848R2 

PLOS ONE

Dear Dr. Yang, 

I'm pleased to inform you that your manuscript has been deemed suitable for publication in PLOS ONE. Congratulations! Your manuscript is now being handed over to our production team.

Kind regards, 

on behalf of

Dr. Ali Mohammad Alqudah 

Academic Editor

PLOS ONE